# Urinary Biomarkers for Lupus Nephritis: A Systems Biology Approach

**DOI:** 10.3390/jcm13082339

**Published:** 2024-04-18

**Authors:** Mohamed H. Omer, Areez Shafqat, Omar Ahmad, Juzer Nadri, Khaled AlKattan, Ahmed Yaqinuddin

**Affiliations:** 1School of Medicine, Cardiff University, Cardiff CF14 4YS, UK; omermh2@cardiff.ac.uk; 2College of Medicine, Alfaisal University, Riyadh 11533, Saudi Arabia; oahmad@alfaisal.edu (O.A.); juzernadri@gmail.com (J.N.); kkattan@alfaisal.edu (K.A.); ayaqinuddin@alfaisal.edu (A.Y.)

**Keywords:** lupus nephritis, proteomics, transcriptomics, metabolomics, microRNAs, biomarkers

## Abstract

Systemic lupus erythematosus (SLE) is the prototypical systemic autoimmune disorder. Kidney involvement, termed lupus nephritis (LN), is seen in 40–60% of patients with systemic lupus erythematosus (SLE). After the diagnosis, serial measurement of proteinuria is the most common method of monitoring treatment response and progression. However, present treatments for LN—corticosteroids and immunosuppressants—target inflammation, not proteinuria. Furthermore, subclinical renal inflammation can persist despite improving proteinuria. Serial kidney biopsies—the gold standard for disease monitoring—are also not feasible due to their inherent risk of complications. Biomarkers that reflect the underlying renal inflammatory process and better predict LN progression and treatment response are urgently needed. Urinary biomarkers are particularly relevant as they can be measured non-invasively and may better reflect the compartmentalized renal response in LN, unlike serum studies that are non-specific to the kidney. The past decade has overseen a boom in applying cutting-edge technologies to dissect the pathogenesis of diseases at the molecular and cellular levels. Using these technologies in LN is beginning to reveal novel disease biomarkers and therapeutic targets for LN, potentially improving patient outcomes if successfully translated to clinical practice.

## 1. Introduction

Systemic lupus erythematosus (SLE) is a multisystem autoimmune disorder with heterogeneous clinical manifestations [1]. It most commonly affects women of childbearing age. Lupus nephritis (LN) describes renal involvement in SLE, affecting approximately 40–60% of patients, with African Americans being at higher risk both of LN development and severe forms of LN [2,3]. Patients with LN usually present with findings of nephritic (e.g., hematuria, generalized edema, hypertension) and/or nephrotic (generalized edema, frothy urine) glomerular disease. Urinalysis often reveals proteinuria and hematuria with red blood cell (RBC) casts. Other routinely performed investigations for LN include renal function tests, 24 h urinary proteinuria or spot urine protein-to-creatinine ratio, complement levels, and serologies to assess for anti-double stranded DNA (dsDNA) and anti-nuclear antibodies (ANAs).

The presence of LN portends a poor prognosis in patients with SLE, associated with a significant morbidity and mortality burden [4]. Despite advancements in immunomodulatory therapies, approximately 30% of patients with LN develop end-stage kidney disease, requiring renal replacement therapy [5,6].

The gold standard technique for diagnosing and monitoring LN remains a kidney biopsy [7,8]. The pattern of glomerular disease on renal biopsy is also used to classify LN into six distinct subtypes. However, a kidney biopsy is an invasive technique with an array of potential complications [9]. Moreover, patients with LN often require repeated kidney biopsies to assess disease activity, which is associated with a compounding risk of adverse events [7,8]. On the other hand, traditional serum and urinary biomarkers such as complement levels, glomerular filtration rate, and urine protein-to-creatinine ratios offer limited specificity. They also cannot distinguish LN from other etiologies of nephritis [10,11,12]. Furthermore, the treatment for LN—corticosteroids and immunosuppressants—targets the immune response, not proteinuria [13]. It is possible for inflammation to subsist despite improving proteinuria.

Hence, a need emerges to identify novel markers that meet the following requirements: they reflect the biology of LN; they can be measured easily, preferably through a non-invasive, routinely collectible sample such as urine; and their temporal changes reflect disease activity in a manner that can reflect a change in clinical state, such as disease progression and treatment response. Thoroughly assessing lupus nephritis requires a protocol that combines novel biomarkers with clinical disease activity scores, renal biopsy findings, conventional laboratory markers such as proteinuria and renal function tests, and imaging tools.

In the era of precision medicine and novel immuno-technologies, multi-omic techniques represent a novel strategy to identify potential biomarkers for LN [14,15]. Omics describes a comprehensive assessment of a particular set of molecules. The most utilized example of a genomics approach is genome-wide association studies (GWASs), for instance, in which the genotype of several thousands of individuals is analyzed for genetic markers, and statistically significant differences in the frequency of a genetic variant between cases and controls are taken as evidence of an association between that genetic variant and disease [16]. The techniques used for transcriptomic, proteomic, and metabolomic analyses—the main aspects of multi-omics discussed in this review—are detailed in their corresponding sections.

This review discusses the application of multi-omics technologies on urine to identify biomarkers for LN that reflect its disease burden/severity and can also be used to monitor treatment responses, thereby providing an alternative to the current gold standard of renal biopsies that are invasive and consequently carry inherent risks and contraindications and can be unfeasible to conduct serially in individual patients.

## 2. Urinary Transcriptomics

Transcriptomics involves the quantitative and/or qualitative analysis of different types of RNA molecules (e.g., mRNA and non-coding RNAs, such as microRNAs) expressed in a given sample. Microarray-based analysis was historically the primary method for global transcriptional profiling. RNA sequencing (RNAseq), either in bulk (i.e., bulk-RNAseq) or at the single-cell level (i.e., scRNA-seq), subsequently emerged and is now increasingly used. ScRNA-seq reveals the gene expression patterns of individual cells in tissues and has revealed a previously underappreciated complexity of cellular transcripts that change in health and disease and are contingent upon the organ of context and tissue microenvironment.

The application of transcriptomics approaches to urine is an emerging approach to identifying predictive biomarkers that better reflect disease severity and treatment response than proteinuria [17,18]. Most studies compare scRNA-seq findings of renal biopsy specimens between patients with LN and controls to identify disease-associated phenotypes in renal parenchymal cells and immune cells that may be candidates for being considered therapeutic targets [19,20].

ScRNA-seq findings of non-lesional, non-sun-exposed skin biopsies of patients with LN and healthy controls showed that upregulated keratinocyte IFN responses could distinguish patients with LN from controls [21,22]. These findings indicate that transcriptomic analyses of a readily accessible site such as the skin could represent a novel biomarker for LN monitoring. In the kidneys, an interferon and pro-fibrotic signature elaborated by renal tubular epithelial cells has been associated with a failure to respond to treatment [22]. Similarly, the expression of several transcripts in serial kidney biopsy pre- and post-treatment related to innate and adaptive immune cell activation, including interferon responses, can distinguish treatment responders from non-responders [23].

### MicroRNAs

MicroRNAs are non-coding RNA molecules that regulate the stability of mRNA, thereby controlling its translation into protein and regulating an array of physiological and pathological processes [24,25]. In the context of LN, several microRNAs contributing mechanistically to various pathophysiological processes in LN—from the modulation of inflammatory responses to pathways related to renal fibrosis—have been identified as potential diagnostic biomarkers and indicators of disease activity (Figure 1) [26].

Both urinary and plasma microRNAs are often quantified non-invasively using a real-time quantitative polymerase chain reaction (RT-qPCR), owing to this method’s high sensitivity, cost, and time efficiency [25,27]. MiRNA-21, a critical mediator of inflammation that upregulates interleukin-6- and NF-κB-associated pathways and modulates lymphocyte signaling, was one of the first identified microRNAs associated with LN [28,29,30,31]. Urinary miRNA-21 could distinguish between inactive and active LN in a cohort study of 55 patients with SLE and 30 healthy controls with an area under the curve (AUC) of 0.89 [32]. Furthermore, in 52 patients with LN, miRNA-21 could distinguish between healthy controls and patients with LN with a sensitivity of 86% and an AUC of 0.91 [33]. Urinary miRNA-146a, also a modulator of the NF-κB inflammatory pathway, could similarly accurately discriminate patients with LN from healthy controls [34,35]. Interestingly, miRNA-146a also correlated directly with disease activity and histological features, indicating its potential to predict disease severity and response to therapy [36]. Finally, miRNA-29c, a key modulator of renal fibrosis, has recently been shown to accurately determine renal chronicity and disease severity in LN with a sensitivity and specificity of over 80% [37,38,39].

Recently, a paradigm shift has emerged in urinary transcriptomics with the development of novel biomarker panels utilizing several microRNAs with differing utilities to accurately aid in diagnosing and assessing disease activity [39,40,41,42]. For instance, a recent panel using three microRNAs (miRNA-21, miRNA-150, and miRNA-29c) was evaluated in a cohort study of 45 patients with LN and 20 controls [39]. This microRNA panel showed that changes in these individual microRNA levels correlated significantly with the LN chronicity index (CI), which was a significant predictor of renal fibrosis recorded by immunohistochemistry. Additionally, evaluating a separate panel consisting of three micro-RNAs (miRNA-135b-5p, miRNA-107, and miRNA-31) in a cohort of 42 patients with LN, comprising 21 responders and 21 non-responders, revealed significant differences amongst responders and non-responders [43]. They could accurately predict disease activity and progression during flare-up periods and one year following the initial flare-up (AUC of 0.73–0.78).

## 3. Urinary Proteomics

Proteomics approaches broadly characterize peptide abundance or interactions in a specific sample. The primary technique applied to proteomic study is spectrometry (MS). Mass cytometry adds more resolution to proteomics, which allows us to identify the proteins expressed by different cell types. The advent of mass spectroscopy also led to large-scale metabolomic analyses aimed at characterizing the abundance of molecules like fatty acids, amino acids, and carbohydrates. Mass spectroscopy imaging has been devised to provide spatial context about the proteins and metabolites present within a sample lost in bulk-level mass spectroscopy.

The application of urinary proteomic techniques to identify biomarker proteins has demonstrated excellent potential across several systemic and renal disorders [44]. Identifying urinary biomarker proteins can facilitate the non-invasive diagnosis of LN while aiding in the characterization of disease activity and severity. Additionally, urinary proteomics can pave the way for identifying biomarkers that predict responsiveness to therapy and propensity toward relapse [45].

Two primary approaches underpin the utilization of urinary proteomics in the context of LN biomarker discovery [46]. Targeted proteomics offers a unique perspective as it involves the study of proteins with an established pathophysiological role in the context of LN. Although limited in scope, this approach allows the identification of biomarkers that have biological credibility as they are directly involved in the immunopathogenesis of LN. On the other hand, untargeted proteomics in the form of unbiased discovery proteomics allows for discovering a wide array of potential urinary protein biomarkers that are upregulated or downregulated in patients with LN.

### 3.1. Pro-Inflammatory Biomarkers

Urine proteomics has revealed that numerous cytokines, including IL-17, TWEAK, and MCP-1, can be elevated in patients with LN and positively correlate with disease activity [12]. Screening 1000 urinary protein biomarkers in 30 patients with LN and correlating the urinary protein signature with a single-cell transcriptomic analysis of renal biopsy specimens, Fava et al. demonstrated a robust link between urinary chemokine signals and renal immune cell infiltration, indicating that this approach may be more reflective of the immune status of the kidney than urinary proteinuria—currently widely utilized [47]. This approach also allowed for the stratification of patients with LN over a gradient of IFNγ-inducible chemokines, thereby adding biologically relevant information beyond traditional histological classifications [47]. Furthermore, these findings indicate the possibility of dynamically tracking LN status non-invasively through urine samples over time in a manner that could guide treatment decisions [47].

Recent urinary proteomic studies have revealed a consistent elevation in IL-16 in the urine of patients with active LN [48]. IL-16, a potent T-cell chemoattractant, significantly correlates with renal activity and the NIH indices—histopathology-based scoring systems designed to quantify the severity and progression of LN [49]. Moreover, an early decline in urinary IL-16 at 3 months correlated with treatment response to immunosuppression—a response determined based on UPCR, serum creatinine, and the dose of prednisone—outperforming traditional measures like UPCR [48]. In addition to its predictive value for treatment response, IL-16 can distinguish between proliferative LN and pure membranous LN with an AUC of 0.89 [49]. Additionally, urinary IL-16 abundance correlated with single-cell RNA sequencing analyses of renal biopsies, indicating that IL-16 is produced by infiltrating immune cells in LN kidneys, supporting its utility as a biomarker for monitoring intrarenal immune activity [49]. In 225 patients with LN, urine proteomic signatures showed that fibrous crescents were similar to activity-related lesions despite being considered inactive lesions [48]. Despite their classification under the NIH chronicity index, an inflammatory signature including CD73, MMP9, MIP1b, and IL-8 was identified in fibrous crescents, highlighting the potential for tailored interventions [50].

Beyond IL-16, CD163—a macrophage-specific hemoglobin scavenger receptor upregulated during inflammation—has been consistently identified as a urinary biomarker through ELISA and single-cell transcriptomics techniques in patients with LN [51,52,53]. Urinary CD163, closely following IL-16, significantly correlates with LN severity indicated by the NIH activity index and histological activity [48,54]. Urinary CD163 concentration improved considerably by week 12 in complete treatment responders, with a decline at three months predicting a one-year response more accurately than proteinuria [48]. Elevated CD163 levels across all LN classes, especially in proliferative forms, were closely linked to disease activity and treatment response, highlighting its utility as a non-invasive marker for tracking LN progression and therapeutic efficacy [48].

In addition to cytokines, a urine proteomic analysis has unveiled numerous signaling molecules as biomarkers of LN pathogenesis [14]. The application of extensive proteomics revealed a panel of six biomarkers (ICAM2, FABP4, FASLG, IGFBP-2, SELE, and TNFSF13B/BAFF) that effectively distinguished (AUC ROC > 0.8) patients with LN and active renal disease (AR) from those with inactive disease (iSLE), with the majority also showing a strong correlation with clinical disease activity [55]. Other promising urine biomarkers—such as Angptl4, L-selectin, TPP1, and TGFβ1—also had high ROC AUC values for distinguishing patients with lupus and AR from those with iSLE, with the combination of Angptl4, L-selectin, and TPP1 yielding the highest discrimination with an AUC of 0.97 [56]. Urinary L-selectin and Angptl4 preceded or coincided with worsening renal disease activity as measured by the renal domains of the Systemic Lupus Erythematosus Disease Activity Index (rSLEDAI), supporting a causal relationship between their elevation and LN severity [56]. However, not all these urinary proteins are suitable for diagnosing LN, as their levels may rise in other conditions causing chronic kidney disease [56]. For instance, Angptl4, L-selectin, and TPP1 were elevated in CKD secondary to numerous cases, TGFβ1 was only increased in FSGS, and urinary Angptl4 correlated with the CKD stage (correlation coefficient: 0.56; *p* < 0.0001) [56]. Additional urinary biomarkers such as ORM1 hold potential for the early detection of LN, even before the onset of significant proteinuria [57]. Another study on 92 patients investigated the use of high-throughput proteomics to identify urine-based markers for tracking kidney disease activity and damage in patients with LN monitored by the NIH activity index (NIH-AI) and chronicity index (NIH-CI) scores [58]. The study identified eight urinary markers (ApoA-II, vWF, IL-1α, IGFBP2, IL-6Rβ, KIM-1, DBH, and fetuin-A) and developed two predictive algorithms with over 88% specificity and 93% accuracy [58]. As longitudinal kidney biopsies are not typically performed for disease monitoring, these urinary markers hold promise for non-invasively tracking changes in LN status over time.

The role of a complement in LN is gaining attention due to the emergence of complement-targeting therapeutics [59]. In a study of 30 patients with LN, the kidney deposition of Membrane Attack Complex (MAC)—the terminal product of complement activation—was positively associated with tubulointerstitial fibrosis and atrophy (IFTA) and proteinuria, which are predictors of progression to ESKD [60]. Urinary proteomic assays in 46 patients with LN demonstrated that patients with more severe IFTA had a higher ratio of C9 to CD59 than those with no/mild IFTA [60]. Urinary complement activation markers also correlated with the increased expression of genes involved in TGFβ and PDGFRβ signaling, indicating a potential link between terminal complement pathway activation in kidney tubules and critical growth factors in developing kidney fibrosis in LN [61,62]. These findings align with transcriptomic data demonstrating that TGFβ1 can stimulate the expression of C3 in the kidneys and with studies showing that PDGFRβ-positive pericytes can secrete complement factor C1q in murine models of renal fibrosis [63,64]. TGFβ has been recognized for its crucial role in activating pro-apoptotic pathways, leading to renal fibrosis in patients with lupus, which can contribute to persistent immune stimulation and epitope spreading, consequently worsening autoimmunity [65].

### 3.2. Rail Score

A combination approach that utilizes both targeted and untargeted proteomics techniques to generate a novel biomarker panel represents an excellent strategy for biomarker development in the context of LN. This strategy has led to the recent development of the Renal Activity Index for Lupus Nephritis (RAIL) [45]. The RAIL score was developed by selecting urinary biomarkers from targeted and untargeted proteomics studies and applying multivariate regression analyses to identify the six most discriminative urinary biomarkers for LN—NGAL, MCP-1, KIM-1, ceruloplasmin, adiponectin, and hemopexin.

The RAIL score represents a novel biomarker panel comprising six urinary protein biomarkers, two from targeted proteomics and four from untargeted proteomic techniques. The two proteins utilized in the RAIL identified via targeted proteomics are neutrophil gelatinase-associated lipocalin (NGAL) and monocyte chemoattractant protein-1 (MCP-1). Mechanistically, NGAL represents a nephroprotective protein consistently upregulated in patients with various forms of renal injury in renal epithelial cells [66]. In contrast, MCP-1 is a chemokine that regulates immune cells’ diffuse infiltration into renal tissue, facilitating the ongoing inflammatory cascade in LN [67]. Targeted proteomic techniques include enzyme-linked immunosorbent assays (ELISAs) to accurately quantify NGAL and MCP-1 in urinary samples amongst patients with LN [68]. Numerous cross-sectional and prospective cohort studies have shown that NGAL and MCP-1 significantly increase in patients with LN [69,70,71,72,73]. Moreover, higher levels of NGAL and MCP-1 were correlated with heightened disease activity, and these biomarkers could predict subsequent relapses and disease flares accurately [74,75,76,77]. The findings of these cohort studies, which jointly included approximately three hundred patients, led to the incorporation of MCP-1 and NGAL into the RAIL score.

In contrast, the other four biomarkers incorporated in the RAIL score were identified using unbiased discovery proteomics techniques, such as mass spectrometry. Four primary proteomic discovery studies pioneered the identification of novel urinary biomarkers in the context of LN [77,78,79,80,81]. These studies compared urine proteomic signatures across approximately 300 patients with LN and controls. They identified ~30 target proteins through the utilization of techniques such as surface-enhanced laser desorption–ionization time-of-flight mass spectrometry (SELDI-TOF) and matrix-assisted laser desorption–ionization time-of-flight mass spectrometry (MALDI-TOF-MS/MS). Identified proteins were then validated in further targeted proteomics studies utilizing ELISA techniques [81,82,83,84]. Subsequently, four of these proteins were selected in the RAIL score, including ceruloplasmin, adiponectin, hemopexin, and kidney injury molecule-1 (KIM-1). Mechanistically, hemopexin and ceruloplasmin represent antioxidant proteins that are increased in subjects with LN, potentially as an indicator of ongoing inflammation [85,86]. Furthermore, KIM-1 is in the nephron and is upregulated during kidney damage to facilitate the clearance of damaged cells [87,88]. Finally, adiponectin suppresses inflammation that can be upregulated in patients with LN [89]. Figure 2 provides an illustrative demonstration of the RAIL biomarkers and their function in the kidney. Importantly, proteins such as NGAL and KIM-1 are not specific for LN and reflect kidney injury secondary to a variety of causes (reviewed here: [90]).

The first study to validate the RAIL score was conducted in a cohort of 46 children and adolescents and demonstrated outstanding efficacy with the capability to identify over 90% of LN cases [91]. Furthermore, the RAIL score outperformed conventional diagnostic scores that utilized traditional serum and laboratory biomarkers. Following this, the RAIL score was validated in a further study using 79 adult patients with LN, and it demonstrated excellent efficacy with an AUC of 0.88, indicating excellent diagnostic capacity [92]. Additionally, a third study set out to validate the predictive accuracy of the RAIL score in predicting response to therapy and disease flares amongst a cohort of 87 patients with LN [93]. In the study, the RAIL biomarkers could accurately predict response to therapy and propensity towards relapse, highlighting that these biomarkers not only hold diagnostic potential but are also capable of predicting the prognosis. RAIL scores could also distinguish clinically active LN from inactive LN or healthy controls in pediatric patients with SLE and decreased by ≥1 point in patients with complete remission [94]. Importantly, the RAIL score outperformed the rSLEDAI in capturing high LN activity (AUC of 0.79 vs. 0.62, respectively). Furthermore, the RAIL score could reveal subclinical/low–moderate LN activity in patients with an rSLEDAI of 0 who had a kidney biopsy more than 3 months ago [95], underscoring its potential as a method to routinely monitor subclinical kidney disease [94].

## 4. Urinary Metabolomics

Metabolomics approaches involve comprehensively profiling low-molecular-weight metabolites in a given biological sample. Nuclear magnetic resonance (NMR) spectroscopy and mass spectrometry are the main methods for conducting metabolomic analyses. Numerous studies have leveraged these techniques to identify novel biomarkers associated with various healthy and disease states, including cardiovascular disease [96], neurodegenerative disease [97], aging [98], pregnancy complications [99], and autoimmune diseases [100].

In this regard, NMR spectroscopy of serum of patients with LN, patients with SLE and without LN, and healthy controls has demonstrated that patients with LN have higher levels of lipoproteins (VLDL and LDL), but lower levels of acetate, than patients with SLE [101,102]. NMR spectroscopy of serum was used to describe that a combination of three metabolites (neuritic acid, C1q, and cystatin-C) could distinguish patients with SLE and LN from those without LN with an AUC of 0.9 [103]. Interestingly, these metabolomic changes are reversed upon treatment with cyclophosphamide-based therapy for 6 months, with LDL/VLDL levels decreasing and acetate levels increasing, with these changes also correlated with SLEDAI, renal SLEDAI, and serum C3 and C4 levels [102]. These findings indicate that serum-based metabolomics can distinguish patients with LN from SLE without LN and from healthy controls, as well as identify differential responses to treatment, which can thereby be used for monitoring therapy response.

Urinary metabolomics approaches in LN have been used to identify metabolites differentially affected in distinct histological classes of LN and monitor treatment responses. Compared to healthy controls, urinary pyruvate, citrate, fumarate, malate, and α-ketoglutarate are significantly decreased in patients with LN [104,105]. Comparing NMR-based metabolomic profiling in seven patients with class III/IV LN vs. class V LN, Romick-Rosendale et al. showed that urinary citrate was significantly lower in class V LN. In contrast, urinary taurine and Hippurate were markedly lower in class III/IV LN than type V LN [106]. In another study of six patients with pure class III/IV LN, seven patients with pure class V LN, and seven with mixed type III/IV + V LN, the ratio of picolinic acid to tryptophan (Pic/Trp) in urine was significantly lower in patients with type V LN than those with class III/IV. Combining the Pic/Trp ratio with eGFR and the urinary protein-to-creatinine ratio (UPCR) could distinguish the LN classification of pure type III/IV vs. pure type V LN with an AUC of 0.91, outperforming eGFR alone (0.499) and UCPR alone (0.444), which are current laboratory measures for monitoring LN [104].

In the context of treatment response, urinary citrate, which was significantly lower in patients with LN than healthy controls and distinguished between them with an AUC of 0.91, increased 6 months after induction therapy with cyclophosphamide for LN [105]. Urinary citrate levels also correlated moderately but significantly with C3 (r = 0.362; *p* = 0.03) and UPCR (r = −0.346; *p* = 0.039). Although urinary acetate levels—higher in patients with LN than healthy controls at the disease diagnosis—did not decrease significantly post-treatment, they did correlate significantly with SLEDAI (r = 0.337; *p* = 0.048). Considering this evidence and serum-based metabolomics studies that identified LN biomarkers, these findings pave the way for monitoring LN treatment response through routine blood samples or non-invasively through urine instead of renal biopsies. However, it remains to be investigated to what degree these treatment-related changes are brought about by clinically beneficial treatment responses or as an independent effect of these medications on the serum and urine metabolome [107].

## 5. Conclusions and Perspectives

Table 1 provides an overview of the transcriptomic, proteomic, and metabolomic biomarkers discussed within this review. Utilizing urinary transcriptomics, proteomics, and metabolomics in LN holds tremendous potential to aid in establishing accurate diagnoses and predicting the therapeutic response and prognosis. It is unlikely that urinary microRNAs and the RAIL biomarkers will replace conventional diagnostic modalities such as kidney biopsies and other traditional biomarkers. However, an approach utilizing urinary microRNAs, the RAIL and other pro-inflammatory biomarkers, and urinary metabolites in combination with conventional diagnostic methods offers a unique perspective toward enhancing diagnostic and prognostic accuracy. Moreover, urinary multi-omic protocols can eliminate the need for multiple kidney biopsies amongst patients with LN, as urinary microRNA, metabolites, and protein biomarkers can predict disease activity, reducing the repeated utilization of invasive procedures. Developing a multi-parameter urinary panel incorporating different types of multi-omic biomarkers could facilitate the generation of a novel, non-invasive modality that integrates emerging immuno-technologies to accurately aid in the disease diagnosis, predict the prognosis and therapeutic response, and assess the presence of chronic disease and renal architectural damage. Finally, such biomarkers can provide novel insights into the pathophysiology of the disease and its mechanisms, leading to advances in the understanding and targeted treatment of the disease.

Nonetheless, several challenges and limitations remain in utilizing urinary transcriptomics and proteomics in the clinical setting. More extensive prospective studies across diverse patient cohorts remain a need to validate the findings of urinary biomarkers, including the RAIL biomarkers and urinary microRNAs. In 368 adolescents and young adults, RAIL biomarkers amongst the healthy controls demonstrated substantial variations in biomarker levels due to age and gender [108]. Moreover, several studies have found conflicting evidence regarding quantifying specific urinary microRNAs in the context of lupus nephritis [109]. Hence, validation studies establishing reference ranges for these urinary biomarkers to distinguish healthy controls from patients with LN accurately are needed.

Additionally, cutting-edge quantitative transcriptomic, proteomic, and metabolomic techniques are required to quantify these biomarkers accurately in the clinical setting. The majority of the studies validating the RAIL biomarkers utilized ELISA techniques for protein quantification; however, given the emergence of novel approaches such as mass spectrometry to quantify proteins, the incorporation of these techniques in the clinical setting is required owing to their accuracy and potential cost-effectiveness when utilized at a grander scale [110]. On the other hand, the majority of studies quantifying urinary micro-RNAs used RT-qPCR to quantify microRNAs; however, the emergence of next-generation sequencing technology offers unique perspectives toward the quantification of microRNAs due to its greater sensitivity and capacity to quantify total microRNA signatures [111]. Additionally, mechanistic research efforts are required to further elucidate the contribution of identified biomarkers towards the pathophysiology of LN. The confounding effect of treatment (including corticosteroids and immunosuppressants), patient comorbidities, and lifestyle factors on urinary transcriptomic, proteomic, and metabolomic findings must also be better understood, as this could majorly affect the interpretation of findings in individual patients. Finally, collaborative efforts are needed between scientists and clinicians to bridge the gap and raise awareness regarding the tremendous utility of these biomarkers in the clinical setting. Metabolomics approaches are increasingly utilized in the field of cardiovascular disease prevention and management in the form of composite metabolomic risk scores or lipidomic scores, which signals increasing clinician awareness of these novel approaches. Nevertheless, more must be done to increase knowledge of these technologies if they are to be introduced widely into clinical care, including the possibility of integrating their basics into the medical education curriculum.

## Figures and Tables

**Figure 1 jcm-13-02339-f001:**
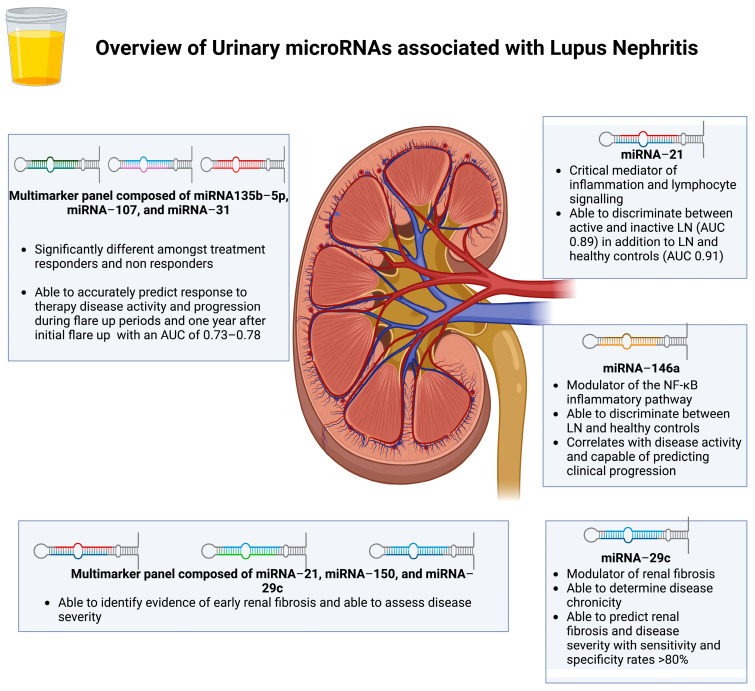
An overview of urinary microRNAs associated with lupus nephritis. This figure was created using BioRender.com.

**Figure 2 jcm-13-02339-f002:**
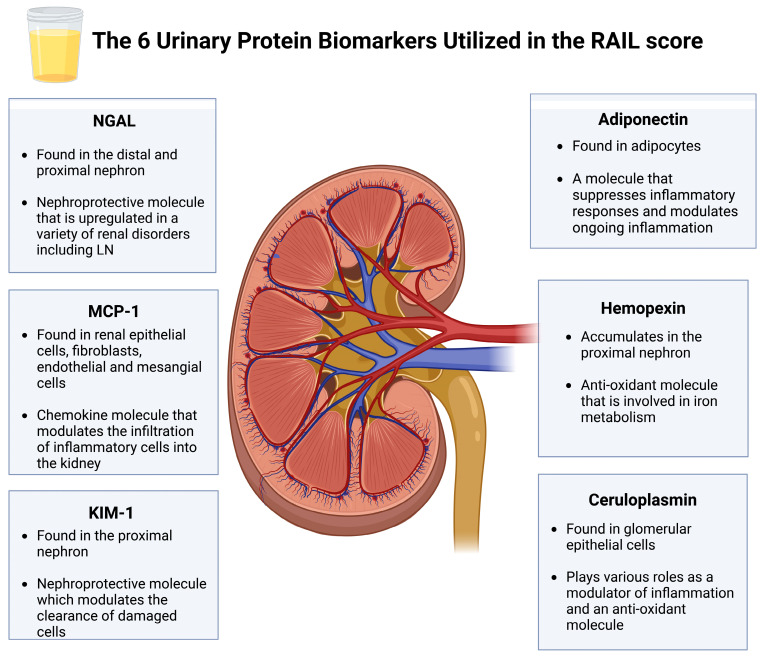
An overview of the components of the RAIL score. This figure was created using BioRender.com.

**Table 1 jcm-13-02339-t001:** Summary of biomarkers for lupus nephritis.

Biomarker	Association	References
Urinary Transcriptomics		
miRNA-21	Distinguished inactive and active LN (AUC = 0.89) and differentiated healthy controls from patients with LN (sens.: 86%; AUC = 0.91).	[28,29,30,31,32,33]
miRNA-146a	Accurately discriminates patients with LN from healthy controls, directly correlates with disease activity and histological features.	[34,35,36]
miRNA-29c	Key modulator of renal fibrosis, accurately determines renal chronicity and disease severity in LN (sens.: >80%; spec.: >80%).	[37,38,39]
MicroRNA Panel (miRNA-21, miRNA-150, miRNA-29c)	Correlated significantly with the LN chronicity index.	[39]
MicroRNA Panel (miRNA-135b-5p, miRNA-107, miRNA-31)	Predicting disease activity and progression, showing significant differences between responders and non-responders during flare-ups and one year after (AUC = 0.73–0.78).	[43]
Urinary Proteomics		
IL-16	Correlates with renal activity and NIH indices, predictive of treatment response, distinguishes LN types with high accuracy (AUC of 0.89).	[48,49]
CD73, MMP9, MIP1b, IL-8	Identifies fibrous crescents.	[50]
CD163	Correlates with LN severity and treatment response, predicts one-year response more accurately than traditional measures.	[48,51,52,53,54]
Proteomic Panel (ICAM2, FABP4, FASLG, IGFBP-2, SELE, TNFSF13B/BAFF)	Distinguishes active renal disease from inactive disease in patients with LN with high accuracy (AUC > 0.8), correlates with clinical disease activity.	[55]
Angptl4, L-selectin, TPP1, TGFβ1	High discrimination power for active vs. inactive disease (highest AUC of 0.97 with a combination), indicates causal relationship with LN severity.	[56]
ORM1	Early detection of LN.	[57]
Urinary apoA-II, vWF, IL-1α, IGFBP2, IL-6Rβ, KIM-1, DBH, Fetuin-A	Identified for tracking kidney disease activity and damage in LN, with high specificity and accuracy.	[58]
Complement Activation Markers (C9-to-CD59 Ratio)	Associated with tubulointerstitial fibrosis and proteinuria, indicators of ESRD progression; linked with TGFβ and PDGFRβ signaling in kidney fibrosis development.	[60,61,62,63,64]
NGAL	Upregulated in renal epithelial cells during renal injury. Significantly increases in patients with LN, correlates with disease activity, and predicts relapses and disease flares.	[66,69,70,71,72,73,74,75,76,77]
MCP-1	Elevated levels associated with increased disease activity and predictive of disease progression.	[67,69,70,71,72,73,74,75,76,77]
Ceruloplasmin	Antioxidant protein increased in LN, potentially indicating ongoing inflammation.	[86]
Adiponectin	Anti-inflammatory protein upregulated in patients with LN.	[89]
Hemopexin	Antioxidant protein increased in subjects with LN, indicating ongoing inflammation.	[85]
KIM-1	Upregulated during kidney damage for the clearance of damaged cells.	[87]
RAIL Score (NGAL, MCP-1, Ceruloplasmin, Adiponectin, Hemopexin, KIM-1)	Diagnostic capability (over 90% identification rate of LN cases in children and adolescents; AUC of 0.88 in adults). Excellent predictive accuracy for response to therapy and disease flares.	[90,91,92,93,94,95]
Urinary Metabolomics		
Serum Lipoproteins (VLDL, LDL), Acetate Levels	Higher VLDL and LDL but lower acetate levels in patients with LN compared to healthy controls. Changes upon treatment correlate with disease activity and treatment response.	[101,102]
Serum Neuritic Acid, C1q, Cystatin-C	Can distinguish patients with SLE and LN from those without LN (AUC = 0.9). Levels reverse upon treatment.	[103]
Urinary Pyruvate, Citrate, Fumarate, Malate, α-Ketoglutarate	Significantly decreased in patients with LN compared to healthy controls.	[104,105]
Urinary Citrate	Significantly lower in class V LN compared to class III/IV. Increases after treatment, correlating with treatment response.	[105,106]
Urinary Taurine and Hippurate	Markedly lower in class III/IV LN than class V.	[106]
Urinary-Picolinic-Acid-to-Tryptophan Ratio (Pic/Trp)	Lower in type V LN than in class III/IV; combined with eGFR and UPCR, distinguishes LN classes with high accuracy (AUC of 0.91).	[104]
Urinary Acetate	Higher in patients with LN at diagnosis; correlates with disease activity (SLEDAI), but does not significantly decrease post-treatment.	[106]

Angptl4: Angiopoietin-like 4; C1q: Complement component 1q; DBH: Dopamine beta-hydroxylase; FABP4: Fatty acid binding protein 4; FASLG: Fas ligand; ICAM2: Intercellular adhesion molecule 2; IGFBP-2: Insulin-like growth factor binding protein 2; IL: Interleukin; IL-1α: Interleukin 1 alpha; IL-6Rβ: Interleukin 6 receptor beta; KIM-1: Kidney injury molecule-1; L-selectin: L-selectin; LDL: Low-density lipoprotein; MCP-1: Monocyte chemoattractant protein-1; MIP1b: Macrophage inflammatory protein 1 beta; miRNA: microRNA; MMP9: Matrix metallopeptidase 9; NGAL: Neutrophil gelatinase-associated lipocalin; ORM1: Orosomucoid 1; SELE: Selectin E; TGFβ1: Transforming growth factor beta 1; TNFSF13B/BAFF: Tumor necrosis factor superfamily member 13B/B-cell activating factor; TPP1: Tripeptidyl peptidase 1; VLDL: Very-low-density lipoprotein; vWF: von Willebrand factor.

## Data Availability

Not applicable.

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
