# Peer review of "Urinary Biomarkers for Lupus Nephritis: A Systems Biology Approach"

_jcm, 2024, doi:10.3390/jcm13082339_

Round 1
Reviewer 1 Report
Comments and Suggestions for Authors
I have reviewed the article titled " Urinary Biomarkers for Lupus Nephritis: A Systems Biology 2 Approach." An interesting study in which the authors investigate
The authors describe urinary markers that could differentiate lupus nephritis from other diseases.
It is a very interesting, well-conceived and well-written work that concentrates all those urinary biochemical markers of relevance to lupus nephritis.
· The authors could mention the physical and chemical characteristics of urine in patients with lupus without nephritis and with nephritis (presence of hematuria, mycoralbuminuria, pH, density, etc.)
· From my point of view, it is well presented and summarized, and what would complete this work would be a table summarizing the urinary biochemical markers in the urine of patients with lupus nephritis and patients with only lupus erythematosus.
Comments on the Quality of English Language
Minor editing of English language required
Author Response
We appreciate your comments and overall positive outlook of our paper. We have addressed your comments below.
- The authors could mention the physical and chemical characteristics of urine in patients with lupus without nephritis and with nephritis (presence of hematuria, mycoralbuminuria, pH, density, etc.)
We have added the following to the introduction:
Lines 31-37: Patients with LN usually present with findings of mixed nephritic (e.g., hematuria, generalized edema, hypertension) and/or nephrotic (generalized edema, frothy urine) glomerular disease. Routine urinalysis often reveals proteinuria and hematuria with red blood cell (RBC) casts. Other routinely done investigations for LN include renal function tests, 24-hour urinary proteinuria or spot urine albumin-to-creatinine ratio, complement levels, and serologies to assess for anti-dsDNA and anti-nuclear antibodies (ANA).
- From my point of view, it is well presented and summarized, and what would complete this work would be a table summarizing the urinary biochemical markers in the urine of patients with lupus nephritis and patients with only lupus erythematosus.
We have added a table summarizing all multi-omic urinary biomarkers for SLE.
Reviewer 2 Report
Comments and Suggestions for Authors
The present article aims to review the actual knowledge about potential urinary biomarkers for Lupus Nephritis. Due to the fact that kidneys are quite frequently involved in lupus and that it has a high degree of morbidity and mortality this is still considered a highly important aspect of systemic lupus erythematosus (SLE) management. The article is nicely and comprehensively written with more than hundred references.
Some remarks:
- Introductory part about lupus and lupus nephritis should be expanded – more information about epidemiology (including some global variations) rows 28-33
- Rows 34-35 please update reference number 7 with last Eular recommendations Fanouriakis A, Kostopoulou M, Andersen J, et al EULAR recommendations for the management of systemic lupus erythematosus: 2023 updateAnnals of the Rheumatic Diseases 2024;83:15-29. Also KDIGO 2024 Clinical Practice Guideline for the management of LUPUS NEPHRITIS should be mentioned.
- Row 52 “s for LN 13,14. -Omics d…” please correct
- Introductory part should discuss more about actual routine possibilities of renal involvement diagnosis. Also lupus nephritis classification should de included and correlations with biomarkers should be discuss
- Please include bibliographic numbers associated with statements of figures 1 and 2
Author Response
Thank you for your comments. We have addressed each below.
- The introductory part about lupus and lupus nephritis should be expanded – more information about epidemiology (including some global variations) rows 28-33
To address this concern, we have added the following to the introduction:
Lines 27-37: Systemic lupus erythematosus (SLE) is a multisystem autoimmune disorder with heterogeneous clinical manifestations 1. It most commonly affects women of childbearing age. Lupus nephritis (LN) is the renal involvement in SLE, affecting approximately 40%-60% of patients, with African Americans being at higher risk both of LN development and severe forms of LN 2,3. People with LN usually present with findings of mixed nephritic (e.g., hematuria, generalized edema, hypertension) and/or nephrotic (generalized edema, frothy urine) glomerular disease. Routine urinalysis often reveals proteinuria and hematuria with red blood cell (RBC) casts. Other routinely done investigations for LN include renal function tests, 24-hour urinary proteinuria or spot urine albumin-to-creatinine ratio, complement levels, and serologies to assess for anti-dsDNA and anti-nuclear antibodies (ANA).
- Rows 34-35 please update reference number 7 with last Eular recommendations Fanouriakis A, Kostopoulou M, Andersen J, et al EULAR recommendations for the management of systemic lupus erythematosus: 2023 updateAnnals of the Rheumatic Diseases 2024;83:15-29. Also KDIGO 2024 Clinical Practice Guideline for the management of LUPUS NEPHRITIS should be mentioned.
Done
- Row 52 “s for LN 13,14. -Omics d…” please correct
This typo has been corrected.
- Introductory part should discuss more about actual routine possibilities of renal involvement diagnosis. Also lupus nephritis classification should de included and correlations with biomarkers should be discuss.
We believe this comment refers to the routinely available tests used to diagnose renal involvement in SLE. To address this concern, we have added the following:
Lines 31-37: People with LN usually present with findings of mixed nephritic (e.g., hematuria, generalized edema, hypertension) and/or nephrotic (generalized edema, frothy urine) glomerular disease. Routine urinalysis often reveals proteinuria and hematuria with red blood cell (RBC) casts. Other routinely done investigations for LN include renal function tests, 24-hour urinary proteinuria or spot urine albumin-to-creatinine ratio, complement levels, and serologies to assess for anti-dsDNA and anti-nuclear antibodies (ANA).
- Please include bibliographic numbers associated with statements of figures 1 and 2
Done.
Reviewer 3 Report
Comments and Suggestions for Authors
Dear authors, with great interest, I am analyzing your manuscript. I have to admit that the work you have done is sufficient. You state 1. (120-122) that This microRNA panel showed that changes in these individual microRNA levels accurately identified early evidence of renal fibrosis and predicted disease severity in these selected patients. I wonder how you prove the evidence of renal fibrosis in these cases. 2. (144-147) that targeted proteomics offers a unique perspective as it involves the study of proteins with an established pathophysiological role in the context of LN. Although limited in scope, this approach allows the identification of biomarkers that have biological credibility as they are directly involved in the immunopathogenesis of LN. Could you name these markers, please? 3. (167-169) moreover, an early decline in urinary IL-16 at 3 months correlated with treatment response to immunosuppression, outperforming traditional measures like UPCR. Do you have the information on which kind of immunosuppression has been used? 4. (201-202) However, not all these urinary proteins are suitable for diagnosing LN, as their levels may rise in other conditions causing chronic kidney disease. Could you present the rate of this rise, please? 5. (272-273) The first study to validate the RAIL score was conducted in a cohort of 46 children and adolescents and demonstrated outstanding efficacy, with the capability to identify over 90% of LN cases. The RAIL score has been used for AKI diagnosis. May you have the parallels with AKI and LN? 6. Finally, from my perspective, it's worth summarizing the most informative evidence-based Urinary Biomarkers for Lupus Nephritis in one table.
Author Response
- (120-122) that This microRNA panel showed that changes in these individual microRNA levels accurately identified early evidence of renal fibrosis and predicted disease severity in these selected patients. I wonder how you prove the evidence of renal fibrosis in these.
To address this comment, we have added the following to our text:
Line 130—132: This microRNA panel showed that changes in these individual microRNA levels correlated significantly with LN chronicity index (CI), which was a significant predictor of renal fibrosis recorded by immunohistochemistry.
- (144-147) that targeted proteomics offers a unique perspective as it involves the study of proteins with an established pathophysiological role in the context of LN. Although limited in scope, this approach allows the identification of biomarkers that have biological credibility as they are directly involved in the immunopathogenesis of LN. Could you name these markers, please?
While we appreciate this comment, we have stated the approach utilized by individual studies after this general introductory paragraph.
- (167-169) moreover, an early decline in urinary IL-16 at 3 months correlated with treatment response to immunosuppression, outperforming traditional measures like UPCR. Do you have information on which kind of immunosuppression has been used?
While the specific medications and doses were not mentioned in the study, we included the criteria the authors used to categorize individual patient response rates, including prednisone dosing.
Line 177-180: Moreover, an early decline in urinary IL-16 at 3 months correlated with treatment response to immunosuppression—response determined based on UPCR, serum creatinine, and dose of prednisone—outperforming traditional measures like UPCR.
- (201-202) However, not all these urinary proteins are suitable for diagnosing LN, as their levels may rise in other conditions causing chronic kidney disease. Could you present the rate of this rise, please?
The study did not report the exact rate of this rise across increasing CKD stages. However, we have expanded the discussion to address this comment:
Lines 213-216: For instance, Angptl4, L-selectin, and TPP1 were elevated in CKD secondary to numerous cases, TGFβ1 was only increased in FSGS, and urinary Angptl4 correlated with CKD stage (correlation coefficient 0.56; p < 0.0001) 4.
- (272-273) The first study to validate the RAIL score was conducted in a cohort of 46 children and adolescents and demonstrated outstanding efficacy, with the capability to identify over 90% of LN cases. The RAIL score has been used for AKI diagnosis. May you have the parallels with AKI and LN?
We appreciate this comment by the reviewer. When introducing the different proteins of the RAIL score, we stated that their levels are altered in various forms of kidney injuries, not just LN. In our opinion, delving into more details about specific comparisons between AKI (which has various forms) and LN may make the section on the RAIL score less focused. However, we have now explicitly stated that markers such as NGAL and KIM-1 are not specific for LN and cited a detailed review covering various markers of kidney injury:
Lines 281-283: Importantly, proteins such as NGAL and KIM-1 are not specific for LN and reflect kidney injury secondary to a variety of causes (reviewed here 5).